# Rereading the Hudaybiyya Treaty: With Special Reference to Ibn ʿUmar's Role in *Fitan* †

**Mursal Farman** [1,*] **and Salih Yucel** [2] 

1    School of Theology, Faculty of Theology and Philosophy, Australian Catholic University, 115 Victoria Parade, Fitzroy, VIC 3065, Australia
2    Centre for Islamic Studies and Civilisation, Charles Sturt University, Bathurst, NSW 2795, Australia
*    Correspondence: mursal.farman@myacu.edu.au or mursal_farman@yahoo.com; Tel.: +61-404854841
†    The article is extracted from a PhD dissertation titled *"Examining Ibn ʿUmar's Stance during Fitan Times and its Impact: Re-reading His Approaches to Peace and Conflict"* that is supported by an Australian Government Research Training Program (RTP) Scholarship.

**Abstract:** The Treaty of Ḥudaybiyya is a brilliant chapter in Islamic history. It can be called *umm muʿāhadāt al-salām* (the mother of peace treaties) in Islamic history. Just as migration to Medina is a dividing line between the periods of religious oppression and political independence for Muslims, Ḥudaybiyya is a boundary between the phases of struggle and domination. The role of this treaty in the spread of Islam was evident from the beginning, and much has been written about it. However, nothing has been produced about the role of ʿAbd Allah b. ʿUmar, inspired by the Ḥudaybiyya treaty, in peacemaking. This paper argues that due to his circumstances, Ibn ʿUmar became the first person to discover the spirit of the Ḥudaybiyya treaty for procuring peace during the *fitan* (civil wars). His efforts were not limited to intellectual achievements, but amid the worst wars of the *fitan*, he tried to practically implement the soul of the Ḥudaybiyya agreement that impacted later generations. He believed that Islam could flourish in a peaceful society, as had happened after the Ḥudaybiyya treaty. The role he played in a tribal society without holding any official position makes Ibn ʿUmar's leadership highly relevant to today's world, where intellectual and spiritual leaders can play a role more pivotal than ever.

**Keywords:** Prophet Muhammad; Ibn ʿUmar; Ḥudaybiyya; peace; *fitan*; companions; Islamic history; Umayyads



## 1. Introduction

The Hudaybiyya treaty is a turning point in Islamic history. Although the treaty's articles were apparently against Muslims, it garnered many fruits. Firstly, it allowed the newly established state in Medina to be recognized by Meccan polytheists. It also became a major stepping stone for its acknowledgements in the region (El Garah et al. 2012). It gave an opportunity to Prophet Muhammad to solidify his position with strategic outcomes (Idris and Sakat 2015). In the same year, the Prophet sent his envoys to Aṣḥama, emperor of Abyssinia; Heraclius, the emperor of the Byzantine; Khosrow, the king of Persia; and Himyarite Ḥārith, the prince of Yemen. By sending envoys, he established diplomatic relations with them, except for the king of Persia. Secondly, the peaceful environment not only gave freedom to Muslims to reach out to their relatives in Mecca but also paved the way to conquer many surrounding tribes. Thirdly, the Prophet's aim was to break up the enemy alliance that besieged Medina in the Battle of the Trench (Avci n.d.). After the Ḥudaybiyya peace treaty, the positions of some Jewish leaders who broke the treaty with Muslims in Medina and of aggressive pagans in the region weakened without Meccan polytheists' support. Finally, it was a great lesson for the companions of the Prophet Muhammad and later generations about the social, spiritual, political, and economic benefits of peaceful coexistence.



It is commonly known that during conflicts and wars, most people cannot judge rationally what is right and wrong or what is true and false (Nursi 1920). With a peaceful environment, Islam flourished in the Arabian Peninsula after the Ḥudaybiyya treaty. More people, particularly noble polytheists such as Khālid b. al-Walīd, ʿAmr b. al-ʿĀṣ and many others, voluntarily discovered and converted to Islam (Sertkaya 2016). By increasing various types of relations with Meccan polytheists and other tribes, Muslims became influential through their moral superiority. Some scholars call the Hudaybiyya treaty a great conquest before the conquering of Mecca (al-Bukhārī 2001, no. 4150; al-Ṭabarī 2000, XXII, pp. 221–22; al-Qurṭubī 1996, XVI, pp. 260–61). Because it was the first treaty of the Muslim state in Medina, it became an inspiration and a model for many peace treaties in Islamic history (Yazir n.d.). That is why the Qurʾān calls it *fatḥ mubīn* (a clear triumph) (Qurʾān, 48:1). Regarding the Hudaybiyya treaty, Said Nursi stated,

> "With the Treaty of Hudaybiyya, Muslims no longer had to resist the attacks of the Makkans with the sword; instead, the brilliant truths of the Qurʾan found a peaceful atmosphere to spread, and it conquered minds and hearts. In this truce, the two sides came to know one another. The virtues of Islam and the light of the Qurʾan rent apart the veils of obstinacy and tribal fanaticism and proved to be very effective." (Nursi 2008, p. 43)

The number of Muslims was less than five thousand until the Hudaybiyya treaty, but two years after the treaty, this figure exceeded ten thousand. The exegetes, when interpreting the Qurʾānic verse that was revealed when the Hudaybiyya treaty was implemented, unanimously agreed that Islam spread rapidly after the treaty. Indeed, in times of peace, Islam spread rapidly. History testifies that Islam flourished in peaceful societies (further discussion on the Hudaybiyya treaty is in Section 3).

ʿAbd Allah Ibn ʿUmar is one of the leading companions who deeply comprehended the spirit of the Hudaybiyya treaty and made it an essential principle for social harmony. To understand the role of this treaty in Ibn ʿUmar's struggle for peace and social harmony, particularly during *fitan* times, it is necessary to shed some light on his life.

## 2. Who Is Ibn ʿUmar?

Ibn ʿUmar (d. 693) was a famous companion of the Prophet and son of the second Muslim Caliph, ʿUmar (d. 644). He was born in Mecca, embraced Islam in childhood, and migrated to Medina with his parents. Poverty in Medina brought him closer to the Prophet, and he stayed at *ṣuffah* (a place in the Prophet's Mosque shaded with palm leaves and used as a shelter for the poor companions of the Prophet) (al-Nīsābūrī 1990, no. 4294; Abū Nuʿaym al-Iṣbahānī 1974, p. 7). His father, ʿUmar, was a prominent member of the Prophet's *shūrā* (council) and very active in the Muslim community of Medina. The experiences of his father played a vital role in his grooming. After his sister Ḥafṣa (d. 665) was married to the Prophet, Ibn ʿUmar also benefited. He was a keen learner, and apart from the aforementioned three, he had the opportunity to see senior companions in critical situations and learn from their conduct. He was a prominent scholar, and his command of traditional Islamic disciplines, including *tafsīr* (Qurʾānic exegesis), *ḥadīth* (prophetic traditions), *fiqh* (Islamic jurisprudence), *sīrah* (the biography of Prophet Muhammad), and *tārīkh* (history), was exemplary. Ibn ʿUmar's life, legacy, viewpoints, and activities are prominently highlighted in the Islamic literature, and his excellence in learning and strict adherence to prophetic practices are acknowledged by various sources (al-Dhahabī 2006, IV, p. 307; al-Shīrāzī 1970, p. 50; al-Qāsimī n.d., p. 72).

The companionship of the Prophet was a period that left a lasting impact on Ibn ʿUmar's personality, and he tried to imbibe *sīrah* (the prophetic way of life) as much as possible. Ibn ʿUmar was almost twenty-one when the Prophet passed away in 632. The sixteen-year-long relationship was the most significant period of his life, and later, the mere mention of the Prophet's name welled his eyes up with tears. His passionate emulation and covetous following of the Prophet were matchless, sometimes making others think of him as crazy or mad (Ibn Saʿd 1968, IV, p. 145; al-Dhahabī 2006, IV, p. 213). While at

home or journeying, he keenly searched for the places where the Prophet had chosen to sojourn or pray (al-Bukhārī 2001, no. pp. 483, 491–92, 505; Yaʿqūb al-Faswī 1981, I, p. 491; Fiṭānī n.d.). He would visit those places to soothe himself after the Prophet's demise and emulate his actions there. He sat under the tree where the Prophet had sat and would not leave until he had watered it, lest it wither away (Ibn al-Athīr 1994, no. 3082). Once, he was in a desert when he remembered the Prophet walking there. He started turning his camel here and there and said, "I wish the hoof of (my camel) touch the place where hoof of the Prophet's camel has hit" (al-Dhahabī 2006, IV, p. 322). He would trim his hair and beard, dye them in the same fashion as the Prophet, and wear similar clothes and shoes. Besides these outward manifestations in his manners, conduct, and worship, he tried to follow the Prophet in letter and spirit (Ibn Saʿd 1968, IV, pp. 105–42; al-Dhahabī 2006, IV, pp. 322–23). All of this shows that his relationship with the Prophet was not limited to his scholarly nature. It was rather an affiliation of the heart, mind, and soul, and this deep association made him a walking and talking source of *sīrah*. Love of the Prophet was so ingrained in his external and internal personality that if any aspect of his life (such as appearance, academic services, social activities, etc.) is studied, it reflects the Prophet's love.

Ibn ʿUmar also ensured the continuity of *siyar* (plural of *sīrah*) of the Prophet. For instance, he named his children after prominent companions of the Prophet (Ibn Ḥanbal 2001, no. 5638). By naming them after the personalities around the Prophet, he not only strove to embody *sīrah* in his life but also endeavored to transfer it to the next generations.

Strictly adhering to the Prophet became such a feature of his character that the next generation watched his every move to learn something new about *sīrah* (Ibn Saʿd 1968, IV, p. 144; Ibn Abī Shaybah 1989, no. 34633). Ibn ʿUmar and *sīrah* became inseparable, so much so that he sometimes had to make clear that certain acts were not prophetic practices (Ibn Saʿd 1968, IV, pp. 154–55).

Ibn ʿUmar's personality led him in a direction opposite to his father's. Just as ʿUmar's rational relationship with Islam and the Prophet played an important role in his legislative wisdom, similarly, Ibn ʿUmar's emotional affiliation with the Prophet was crucial to his spiritual insight. If diversity in personalities compelled ʿUmar to cut down the Ḥudaybiyya tree, it obliged Ibn ʿUmar to pray in places associated with the Prophet and imitate his acts. The father carried out important services in Islamic jurisprudence and administration, whereas the son escaped from public positions due to his specific circumstances and lived his life at the grassroots level. Thus, if the earlier personality influenced laws regarding leadership, the latter impacted legislation about the responsibilities of an ideal citizen. From the perspective of *sīrah*, ʿUmar implemented in his caliphate the lessons he had learned from the leadership of the Prophet, and consequently, his reign became an expansion of the Prophet's era. Likewise, Ibn ʿUmar followed the lessons he received during the reigns of the Prophet and the first two caliphs and called others to observe the same. Therefore, his role at the grassroots level in the times of *fitan* was an extension of the role of an ideal citizen during the life of the Prophet (see Section 5). Both influenced their societies in different ways, and Ibn ʿUmar's impact was never less than his father's. The same has been stated by jurist Abū Salama b. ʿAbd al-Raḥmān b. ʿAwf (d. 713): "ʿUmar lived in a time when he had counterparts, however, Ibn ʿUmar had no equivalent in his age" (al-Shīrāzī 1970, p. 50).

The continuity of the Prophet's *siyar* through Ibn ʿUmar was not restricted to physical practices; he carried it on in key state affairs. A few examples will be mentioned here. The Prophet used to gather important figures of Medina to discuss critical issues (al-Bukhārī 2001, no. 4757; al-Nīsābūrī n.d., no. 1779). Ibn ʿUmar gave the same advice to warring groups before the battle of the Camel (al-Tamīmī 1993, p. 118; al-Ṭabarī 1967, III, pp. 446–51; Ibn al-Aʿtham 1991, II, pp. 452–53). Insincere advice was considered hypocrisy (*nifāq*) in the close circle of the Prophet, and Ibn ʿUmar mentioned the same when ʿUrwa b. al-Zubayr asked him about endorsing the wrong words of the nobles in the caliph's court (al-Faswī 1981, I, p. 377). Examples of senior companions choosing the first and second caliphs influenced Ibn ʿUmar's role in the selection of the third caliph, ʿUthmān (al-Ṭabarī 2000, IV, pp. 220–40). He gave the same advice to Muʿāwiya and stressed public consensus for

the establishment of the caliphate during his multiple meetings before the second wave of *fitan*, as is discussed in Section 5 (Ibn al-ʿArabī 1998, pp. 225–26; al-Ḥasan 2016, p. 573). The Treaty of Ḥudaybiyya is another example of how Ibn ʿUmar's love for the Prophet helped him to continue *siyar*.

### 3. The Ḥudaybiyya Treaty and Ibn ʿUmar's Role

The Peace Treaty of Ḥudaybiyya is an important event in the life of the Prophet of Islam that teaches intellectual, behavioral, and verbal instructions for peace negotiations (Shub et al. 2020). Since their migration to Medina, Muslims were constantly at war with the pagans of Mecca, and many bloody battles took place. However, the Prophet sought opportunities to decrease tension and conflicts between Meccan polytheists and Muslims in Medina. He was in contact with some individuals and observed closely the situation in Mecca. Before the Hudabiyya peace treaty, the Prophet tried to reconcile with the Meccans. He sent a donation of 500 dinars to help the poor of Mecca, who suffered from famine.

> "Afterwards, he married Umm Habiba, the daughter of the chief of the city, Abu Sufyan, He also sent Abu Sufyan plenty of Medina dates and received skins that he could not sell in return. Thus, despite the continuing state of war, he made conciliatory, even friendly gestures on his part, and tried to establish good relations". (Hamidullah 1998, p. 298)

The Hudaybiyya agreement of 628 is also significant because it eased the tension between the two sides; they agreed to a ten-year ceasefire, and the Muslims of Medina peacefully performed a lesser pilgrimage (ʿumrah) the following year.

The Ḥudaybiyya incident began after the Prophet told his companions about a dream of his circumambulation (*ṭawāf*) of the Kaʿba. Since the emigrant companions belonged to Mecca and were revered for the holy shrine, everyone was very happy. The Messenger left with his fourteen hundred companions with the intention of ʿumrah. All of them peacefully reached Mecca in a state of *iḥrām* (a sacred state to perform *ḥajj* or ʿumrah), carrying their sacrifices without weapons, and camped outside the city.

The Quraysh wanted to stop the pilgrims from a rival state of Medina; however, that was against Arab tradition (Ibn Ḥibbān al-Bustī 1996, II, pp. 281–82; Armstrong 2006, pp. 168–69). Ambassadors were exchanged, and the parties agreed to settle the issue through diplomacy rather than warfare. As a result, a ten-year ceasefire agreement was signed on the following conditions: the Muslims would return to Medina that year and would perform pilgrimage in the upcoming year; everyone would be allowed to enter into a covenant with Mecca or Medina, and aggression against anyone would be considered aggression against the party he had joined; and whoever would take refuge with Muhammad without the permission of his guardians would be sent back to Mecca, but not otherwise (Ibn Hishām 1955, II, pp. 317–19; The Editors of Encyclopaedia Britannica 2022). These conditions seemed favorable to the pagan Quraysh of Mecca, and they took it as a victory. On the other hand, the terms were unexpected for the Muslims, and they identified the treaty as a defeat and felt unhappy. A few leading companions had reservations about these terms and expressed dissatisfaction, but could not realize the agreement's significance. Consequences proved it to be the right decision, and the truce opened the way for reconciliation (Rodinson 1985, pp. 250–52).

When the agreement tore down the veil of bigotry, hatred, and rivalry, the Meccans began to discuss the new religion rationally (Armstrong 2006, p. 178). They also realized familiarity and a blood relationship among the crucial figures in the state of Medina. Similarly, the Muslims in Mecca who had not made their way to Medina gained moral success, and their proselytizing activities intensified. Those who were hiding their faith came out more openly, and for those who had adhered to ancestral paganism, belief in idols was now shaken. The political and economic decline also gave the Meccans a different perspective on Islam, the religion of the rising state of Medina. When the Prophet performed ʿumrah the following year along with his companions and spent three days in Mecca as per the agreement, it moved the Meccans and forced them to think. Soon after the Muslims'

return, Khālid b. al-Walīd stood in a gathering of the nobility of the Quraysh and stated, "It has (now) become clear to every wise man that Muhammad is neither a magician nor a poet (as was claimed by us) and his *kalām* (the Qurʾān) is (revealed) from the Lord of the World. It is incumbent upon every man of understanding to follow it" (al-Wāqidī 1855, p. 400). The situation was changing so quickly that ʿIkrima b. Abī Jahl (d. 634) predicted, in these words, "I fear that in less than a year all Meccans will follow (Khālid) and turn Muslims" (Khaṭṭāb 2001, pp. 314–15). The prediction came true, and within a year, important figures of Mecca accepted Islam; this center of paganism for centuries was peacefully conquered in 630.

Ibn ʿUmar was only seventeen at the time of the Ḥudaybiyya agreement, and his role was less important than that of the senior companions. Since ʿUmar was very dynamic, the father's experiences also contributed to his maturity alongside his own observations. On one occasion, when the Prophet sent ʿUthmān b. ʿAffān (d. 656) as an emissary to Mecca and rumors of his martyrdom spread, the Pledge of Pleasing God (*bayʿat al-riḍwān*) was made (Muir 1923, p. 358). The companions scattered to shade themselves under trees before the pledge. When they gathered around the Prophet, ʿUmar sent his son to seek information about new developments. Ibn ʿUmar came to the Prophet and swore allegiance. According to some historians, this made him the first person to pledge allegiance on this occasion (Ibn Qutaybah al-Dīnawarī 1992, p. 162).[1] Then he came along with his father and pledged a second time, thus acquiring the peculiarity of becoming the foremost to swear allegiance and the only companion to pledge twice (al-Bukhārī 2001, no. 4186–87; al-Nīsābūrī 1990, no. 6368). This must have increased the importance of peace for him, and he must have pondered over the wisdom behind it. This allegiance of the Muslims frightened the polytheists and demanded a truce and peace (Atmaca 2021). Ibn ʿUmar reported that when ʿUmar came, he called out to the remaining people for allegiance (al-Wāqidī 1989, II, p. 604).

The young Ibn ʿUmar must have been with his father ʿUmar when he first went to Abū Bakr and then to the Prophet and objected to the peace terms. He must have observed the disruptive reaction of ʿUmar and the composed attitude of Abū Bakr (al-Wāqidī 1989, II, p. 604). The consequences of Ḥudaybiyya and later ʿUmar's regret toward his behavior (al-Wāqidī 1989, II, pp. 606–7) would have taught him to act patiently and courageously on such occasions. Similarly, he must have learned from the successful results of the treaty and the Prophet's example of planning an important venture so secretly that even his closest associates were unaware of it.

After the Ḥudaybiyya treaty, the Prophet instructed the companions to shave their heads.[2] They were in despair and did not move. Then, on the advice of his wife Umm Salama (d. 678) (Adeyemi 2019), the Prophet shaved his head without saying anything. The companions could not resist anymore, and all followed the Prophet's example. Some shaved like him, while others trimmed their hair, as a head-shave would have been performed after ʿumrah, which they had not performed (Armstrong 2006, pp. 175–76). Ibn ʿUmar was present too, and he narrated the Prophet's words about these two groups:

> "May Allah have mercy upon those who have got their heads shaved." The (companions) said: "Messenger of Allah, (what about) those who got their hair clipped?" He said: "May Allah have mercy upon those who have got their heads shaved." They said: "Messenger of Allah, (what about those who have got their hair clipped)?" He said: "May Allah have mercy upon those who got their hair shaved." They said: "Messenger of Allah, (what about) those who got their hair clipped?" He said: "(O Allah, have mercy upon) those who got their hair clipped" (al-Nīsābūrī n.d., no. 1301).[3]

Ibn ʿUmar was too young to fully comprehend the philosophy behind the words of the Prophet at the time of this incident, but later, he must have reflected on and analyzed these experiences. For instance, the Prophet had kept his plan hidden so the companions could not reach any reasonable explanation. The situation made them so disheartened that they disregarded the head-shave. The Prophet took symbolic meaning from the actions of

the two groups (shavers and trimmers). Those who shaved like the Prophet supported him unquestioningly, although they had not appreciated the depth of the matter. In contrast, the actions of those who clipped their hair were below standard. Those people expressed displeasure or hesitance about the decision, as reported by Ibn Mājah (Ibn Mājah n.d., no. 3045).

Ibn ʿUmar reported other traditions about the treaty of Ḥudaybiyya too, showing his attachment to this event and its influence on his personality. For instance, narrations about the disappearance or cutting down of the tree whereunder *bayʿat al-riḍwān* had taken place (Ibn Saʿd 1968, II, pp. 100, 150; Ibn Abī Shaybah 1989, no. 7545), and also about the period for which the treaty could last (al-Nīsābūrī 1990, no. 2354).

### 4. Ibn ʿUmar and Late Converts

A special mention of late converts is required here, as they are closely associated with both of the events relevant to this study: the treaty of Ḥudaybiyya and the civil wars of *fitan* wherein Ibn ʿUmar played his peace-promoting role. Before their conversion to Islam, late converts were the cause of migration of early Muslims from Mecca and had fought against them after their resettling in Medina. The Ḥudaybiyya treaty quickly changed everything, causing mass conversion of the Meccans. Because of their late Islam, circa the conquest of Mecca in 630, the junior companions had not achieved religious or moral maturity, a change brought by Islam; however, the Prophet appointed them to various positions. Ibn ʿUmar was aware of this generation gap in his association with them. We will explain this point with two examples.

Khālid b. al-Walīd was a late convert; he embraced Islam in 630, a year before the conquest of Mecca. Ibn ʿUmar participated in some military campaigns under his command in 630. Due to religious immaturity, Khālid ordered the killing of innocent people. Ibn ʿUmar and other senior companions refused to obey these orders. The Prophet praised this refusal upon receiving the news (al-Bukhārī 2001, no. 4339). Similarly, during the reign of Abū Bakr (632–634), Ibn ʿUmar was again one of the troops under the leadership of Khālid and was sent in 632 toward Ṭalḥa b. Khuwaylid of Banū Asad, northeast of Medina (al-Ṭabarī 1967, III, pp. 248–49), then later dispatched toward Mālik b. Nuwayra of Banū Tamīm at Buṭāḥ. Khālid killed Mālik and sought his widow in marriage. The stance of Ibn ʿUmar and Abū Qatāda al-Anṣarī (d. 676), who had expressed their displeasure of the incident, became famous. Both declined invitations to the subsequent wedding. Abū Qatāda returned to Medina and pledged never to take part in any battle under Khālid, whereas Ibn ʿUmar remained with the troops. Moreover, he advised Khālid to seek permission from the caliph before tying the nuptial knot, but Khālid declined to do so. Ibn ʿUmar's discretion and wisdom later won the approval of Abū Bakr, who ordered an immediate separation between Khālid and the woman (Ibn Khallikān 1900–1994, VI, pp. 14–15; Ibn Ḥajar al-ʿAsqalānī 1994, VI, p. 560).

Khālid was one of the late converts. He was brave, earnest, and hardworking. However, such mistakes were the result of his lack of edification. He was criticized and held accountable by the Prophet but was given another chance after forgoing earlier mistakes. After the Prophet's demise, the senior leadership of the companions (such as Abū Bakr and ʿUmar) had differences over maintaining Khālid or dismissing him from commandership.

Ibn ʿUmar suffered the same fate as the senior companions did at the hands of these late converts before their conversion and had participated in various battles against them. He witnessed the Prophet's treatment of the Meccan Quraysh (the late converts) during the treaty of Ḥudaybiyya and joined military expeditions alongside these recent converts. The two examples above prove the practical experience Ibn ʿUmar gained through coworking with late converts. He must have observed their mistakes and the treatment they received from the Prophet closely. Perhaps all of this influenced his attitude in dealing with the *fitan*. Later, some members of the late converts emerged as opponents of the senior companions on political grounds in the early Islamic civil wars known as *fitan*. Therefore, it is neces-

sary to introduce the *fitan* to provide historical background and a clearer picture of Ibn ʿUmar's efforts.

### 5. *Fitan* and Ibn ʿUmar

The *fitan* occurred between two Muslim groups over the caliphate after the assassination of the third Muslim Caliph, ʿUthmān. The conflict caused two waves of civil wars from 656 to 693. The first wave (655–662) included the battles of the Camel, Ṣiffīn, and Nahrawān, whereas the second wave (680–693) included events such as the martyrdom of Imām Ḥusayn (d. 680), the revolt of Mecca and Medina against Yazīd, the incident of Ḥarra, and other revolts in the Muslim world against the Syrian-based Umayyads (al-Ṭabarī 1967, IV, VI). These wars seem to have been a conflict of approaches (theological, sociopolitical, and tribal) toward the Caliphate.

Apart from the Camel, these battles were between two Muslim groups, each led by senior and junior companions in the first wave and their descendants in the second. The senior companions and their descendants wanted to establish the caliphate in accordance with the will of the people, following the footsteps of the Prophet and the first two caliphs. On the other hand, the other group was led by the Umayyads, who could not run the caliphate according to the standardized mode of the Prophet and the Rashidun caliphs. They held the caliphate by virtue of the organized Syrian army and administration. Their style of governance was tribal and similar to monarchy (Farman 2022, pp. 2–164).

The *fitan* battles were a tragedy in Islamic history and caused thousands of human casualties (al-Ṭabarī 2000, V, p. 539). However, most companions held a different perspective about the battles than did the general public. According to some reports, the total number of companions living in this catastrophic period was about ten thousand, but less than one hundred companions (i.e., one percent) took part in these battles (al-Khallāl 1989, p. 728). Other reports have suggested that their number hardly reached thirty or forty (less than one percent) (al-Khallāl 1989, p. 728; al-Azdī 1983, no. 20735; al-Nīsābūrī 1990, no. 8358). A recent comprehensive study by Fuʾād Jabali concluded that the number was 167, maintaining the fact that a very small minority was involved in the *fitan* (Jabali 1999, pp. 218, 243–44). This consensus of the *ṣaḥābah* upon non-participation in political conflicts later provided the basis for the prohibition of rebellion among *Ahl al-Sunnah*.

The majority of the *ṣaḥābah*'s stance is mentioned above, but Ibn ʿUmar won special distinction among them on various grounds. This does not mean he was superior to other companions; rather, owing to his circumstances during the *fitan*, he had a greater opportunity to try to establish peace.

The first element was Ibn ʿUmar's historical placement among the companions. This meant he had witnessed early Islamic history in Mecca and Medina and also both waves of *fitan* events. Due to his longevity, he was an eyewitness to the ordeals of early Muslims as a youth and later saw battles of unrest, as a mature senior citizen. Most of the senior companions had already passed away by that time, and there was no one else among the emigrant *ṣaḥābah* except him who was present in the second wave of *fitan* and the crushing revolt in Mecca against ʿAbd al-Malik.[4] This historical placement gave him the privilege to witness and participate as a negotiator in most of the *fitan* incidents, in addition to the early Islamic history that made his stance relatively more influential.

A second aspect is that Ibn ʿUmar was not only a pupil of senior companions and taught by them, but also one of them and their companion because he had been part of the Meccan and early Medinan periods of *sīrah*. This gave him an edge among the junior companions, especially those involved in the second wave of *fitan*. To the juniors, he was their companion-cum-mentor. He was an eyewitness to incidents the junior companions did not see.

The third aspect is related to the geographical significance of Ibn ʿUmar's residence. After emigrating to Medina in childhood, he spent the rest of his life there. Here, ʿUthmān's martyrdom took place (Abū al-ʿArab 1984, I, p. 82; al-Dhahabī 1993, V, p. 458) and ʿAlī set off for the battle of the Camel (al-Ṭabarī 1967, III, pp. 446–51; Ibn al-Aʿtham 1991, II,

pp. 452–53). Ibn ʿUmar was present in Medina when the city rebelled against Yazīd and a subsequent massacre took place (Ibn al-ʿArabī 1998, pp. 225–26). He was also present during most of the *fitan* incidents that took place outside Medina. For example, he was there in the arbitration that occurred between ʿAlī and Muʿāwiya at Dumat al-Jandal[5] (al-Dhahabī 2006, IV, p. 316), and also in the solidarity agreement between Imām Ḥasan and Muʿāwiya near Baghdad (Ibn Saʿd 1968, no. IV, p. 182). He tried to stop Imām Ḥusayn in both Mecca and Medina (Ibn al-Aʿtham 1991, V, pp. 25–26). Moreover, he was in Mecca when the second wave of *fitan* events broke out between Ibn al-Zubayr and ʿAbd al-Malik (Ibn al-Athīr 1997, III, p. 405).

The fourth aspect is Ibn ʿUmar's tribal affiliation, which was not from either of two of the most influential branches of the Quraysh (the Hashemites and Umayyads). This had numerous consequences. For instance, Ibn ʿUmar was less swayed by tribal influence, which enabled him to view political issues objectively and with superior understanding. His objectivity was highlighted, as he had kept himself strictly away from political and administrative positions and served voluntarily in the nonpolitical arena. His selfless service, insightful knowledge based on deep acuity, and hands-on experience made him a unique personality of his time.

A fifth aspect is that Ibn ʿUmar's non-participation in the *fitan* battles was not such that he became completely detached from the events of the war as Saʿd b. Abī Waqqāṣ (d. 674) or other companions had done. On the contrary, he took part in all of the events and strove to end war or replace violence and bloodshed with peace and harmony. In other words, he was not passive but proactive during all of these events. Thus, many of his activities, performances, and reactions were recorded. Therefore, although there was a consensus of companions about non-participation in the *fitan* wars, Ibn ʿUmar enjoyed a leading role in peacekeeping.

## 6. Ibn ʿUmar Following Ḥudaybiyya Philosophy in the *Fitan*

The *fitan* caused killing of innocents, created divisions, gave rise to pre-Islamic tribal partisanship, and weakened Islamic leadership. This period coincided with the second half of Ibn ʿUmar's life. During his childhood, he had faced the intolerant attitude of the Meccans toward the Muslims, then seen how the treaty of Ḥudaybiyya turned circumstances in favor of Islam. This strengthened his belief that Islam could flourish in a peaceful society. He knew through experience that if these wars continued, they would consume the energy of the Islamic state, reduce the spread of Islam, stop conquering of new lands, and give foreign powers opportunities to attack Muslims. Political instability would also cause spiritual decline among Muslims and impact rational and positive thinking in politics. These things were on his mind when he opted for an approach distinct from the prevailing approaches. During this period, Ibn ʿUmar acted as an intermediary and conducted dialogues between Caliph ʿUthmān and protestors. He offered judicious advice to ʿUthmān to assign ʿAlī the job of negotiation with the protestors (Ibn al-Aʿtham 1991, II, pp. 409–10). Ibn ʿUmar did not take part in any of these battles but advised both parties, before the battle of the Camel, to consult with the influential figures of Medina. His participation in the arbitration (*taḥkīm*) between ʿAlī and Muʿāwiya and the conciliation (*ṣulḥ*) between Imām Ḥasan and Muʿāwiya highlight his peace-loving personality. In the reign of Muʿāwiya, Ibn ʿUmar participated in at least three meetings with him on Yazīd's nomination for caliphate that later caused the second wave of *fitan* (Khalīfah 1977, pp. 213–17; Ṣafwat n.d., II, p. 248). He also tried to stop rebellions in Medina and Mecca, which were mercilessly crushed, and to dissuade Imām Ḥusayn from leaving for Kufa, which resulted in Imām Ḥusayn's tragic martyrdom. He followed the same model for almost four decades. His endeavors for peace and reconciliation were comprehensive and extended to people from all walks of life. Ḥusaynī and Dafrūr (2020) have suggested that Ibn ʿUmar's peace-promoting activities were the result of what he had learned from the Prophet: not to shed the blood of the Muslims. We agree with this point but have added more elements that played important

roles in the formation of Ibn ʿUmar's personality and led him to assume this direction during the *fitan* period.

As mentioned in Section 5 (lines 346–93), in the second wave of *fitan*, Ibn ʿUmar was the only person who had witnessed the Meccan and early Medinan periods and was also present in Ḥudaybiyya to see how the Prophet peacefully used this opportunity to enter his enemies into the folk of Islam (see Section 1). We have discussed in Section 4 the personal experiences of Ibn ʿUmar and how he was closely connected with the differences of senior companions (such as Abū Bakr and ʿUmar) over leadership of later converts. We also discussed (in Section 5) that excepting in the battle of the Camel, the two rival sides in the *fitan* were first led by senior and junior companions and then by their descendants. As mentioned in Section 2 (lines 91–156), Ibn ʿUmar's heart was attached to the Prophet in such a way that any incident would turn into a mirror in which he could see the reflection of a similar happening in the life of the Messenger. This devotion had converted the Prophet for him to a role model (leader) who he would follow in all respects, minor or major. Even in his senility, he looked for solutions to the evils of the *fitan* in light of *sīrah*. In this regard, he made a conscious effort and hoped for positive consequences.

In 688, Ibn ʿUmar performed *ḥajj* in the presence of multiple Imāms (spiritual leaders to lead the *ḥajj*) who represented multiple conflicting caliphs (Ibn Kathīr 1988, VIII, p. 324). People were divided into followers of four different Imāms and feared that this rift and clash might lead to the cancellation of the *ḥajj* that year. Despite this tense and uncertain situation, Ibn ʿUmar came to Mecca from Medina, saying that if he was not granted permission for *ḥajj*, he would do the same as the Prophet had in Ḥudaybiyya (al-Bukhārī 2001, no. 1708). This is not the sole example; he alluded to the treaty again in 693, when he was barred from performing *ḥajj* on the pretext that a battle was expected. Ibn ʿUmar again rationalized from the incident of Hudaybiyya that if the people were not allowed to perform ḥajj, he would do the same as the Prophet had in Hudaybiyya (Ibn Kathīr 1988, V, pp. 143–44).

His dissuading others from taking part in the protracted battles of *fitan*; participating in every peace-promoting activity; and then referring to Ḥudaybiyya during these wars revealed that behind Ibn ʿUmar's constructive role was the thought of reviving the prophetic peace treaty of Ḥudaybiyya. However, to date, this event has been studied from a jurisprudential point of view alone: that if permission to make a pilgrimage was not given, Ibn ʿUmar would leave his *ḥajj* incomplete in accordance with the Prophet's practice in Ḥudaybiyya. To us, the significance of the mentioned incident goes beyond the boundaries of Islamic jurisprudence but requires a close comparison with the treaty of Ḥudaybiyya. First, both events (the treaty of Ḥudaybiyya and Ibn ʿUmar's reference to it during the *fitan*) took place in states of war. Second, the leadership was almost the same. In the first wave of *fitan*, the two rival factions were led by senior and junior companions, whereas in the second wave, they were led by the descendants of the same leadership. This was very similar to Ḥudaybiyya and worthy enough to remind Ibn ʿUmar of the situation in that treaty. Third, Ibn ʿUmar's role in peace calls for a similar interpretation of this event. Ibn ʿUmar was a figure who had spent four decades of his life advocating for establishment of peace. Was it by chance that only he was destined to recall Ḥudaybiyya amid *ḥajj* days while he was busy asking people to stop fighting? This prompted us to mention Louis Pasteur's (d. 1895) words here: "Chance favors only the prepared mind" (Welter and Egmon 2006, p. 1). The quote, in this context, means that Ibn ʿUmar's peace-promoting activities prepared him to relate Ḥudaybiyya to the *fitan*. Fourth, just as many companions were not pleased with the Prophet's agreement to the treaty's terms, Ibn ʿUmar was also criticized, and many people were not happy with his efforts for peace (Farman 2022, pp. 129–32, 155–57). Fifth, Ibn ʿUmar must have remembered the phrase "manifest victory" used for the Ḥudaybiyya pact in the Qurʾān (48: 1). This phrase was used for the Prophet's accommodation of dealing with the Quraysh and its future consequences. Therefore, Ibn ʿUmar must have considered accommodating a tribal mindset to win rule in keeping with the sunnah of the Prophet at Ḥudaybiyya, wherein the Prophet acceded to the apparently highly unacceptable conditions of the polytheists. Sixth, Ibn ʿUmar understood that despite

being in the right and displeasing the companions, an apparently submissive strategy of the Prophet in Ḥudaybiyya had enabled him to win what could not be won with war. Similarly, Ibn ʿUmar harshly criticized his fellow religious segment,[6] braced their unhappiness, and stopped them from an armed conflict (Farman 2022, pp. 143–50). His strategy reaped advantages that the *fitan* battles, despite of countless sacrifices, could not attain.

Both sides later accepted Ibn ʿUmar's views. The descendants of the senior companions who participated in the *fitan*, for example, Muhammad b. Ḥanafiyya (d. 700), the son of Caliph ʿAlī; Imām Zayn al-ʿĀbidīn ʿAlī b. Ḥusayn (d. 713), the son of Imām Ḥusayn; ʿUrwa b. al-Zubayr (d. 713), the son of al-Zubayr b. al-ʿAwwām; Mūsā (d. 722), the son of Ṭalḥa b. ʿUbayd Allāh; etc., were influenced by him. Similarly, a religious transformation appeared within the Umayyad caliphs that peaked in the reign of ʿUmar b. ʿAbd al-ʿAzīz, wherein the style of government reverted to the Prophet's and the Rashidun Caliphs' ways. Ibn ʿUmar's efforts were then recognized by great scholars in the generation of *ṣaḥābah* (the companions), *tābiʿūn* (the successors), and *tābiʿū al-tābiʿīn* (followers of the successors) (Farman 2022, pp. 168–95). Through these personalities, Ibn ʿUmar left a lasting impact on mainstream Muslims, *Ahl al-Sunnah wa al-Jamāʿah* (the majority, who represent 85% to 90% of the whole Muslim world) (Denny 2011, p. 3), about *fitan*, and the share of his influence is more than of any other companion on the subject (Farman 2021, 2022, pp. 196–212). In this regard, we quote the words of a prominent scholar, Sufyān al-Thawrī (d. 778): "In the time of unity, we take the word of ʿUmar while in the time of division we follow the word of his son, Ibn ʿUmar" (al-Khallāl 1989, p. 138).

Many sources acknowledge Ibn ʿUmar's role in establishing peace, but this paper increases the significance of said role by unearthing new aspects. In the early centuries of Islam, there were some gaps in role models of peace after the Prophet. Therefore, a link between Ibn ʿUmar's efforts for reconciliation and the Ḥudaybiyya treaty may fill this void as to how the companions continued to follow the Prophet's example to promote peace and coexistence.

The companions' affiliation with the Prophet and their roles as narrators of the Qurʾān and *ḥadīth* made their lives sacred. Their actions became a practical interpretation of the holy scriptures crossing the boundaries of history to develop into a part of *dīn* (religion). This gave them an edge over non-companions and elevated their status from local to global. Thus, once it becomes a historical fact that any companion adopted peacekeeping as his full-time job, it gains further importance for Muslims worldwide, especially when violence and religious extremism have become a social evil.

Ibn ʿUmar is particularly relevant to contemporary scholars. He performed his services to peace as a learned scholar and a common citizen without accepting any position. In a tribal society, he influenced the public and the state alike, as do NGOs in today's era of democracy. In today's age of intellectual leadership, scholars have more opportunities than ever to address societies positively. For them, Ibn ʿUmar offers an example of bringing about social change by keeping peace and neutrality.

Ibn ʿUmar's strategy, inspired by the Hudaybiyya treaty, involved three major principles. The first was to sacrifice many things, including any positions, worldly ambitions, and tribalistic politics for the sake of peace in society. He preferred peaceful resistance for social harmony over position, any worldly ambition and tribal or individual honor. The second was using diplomacy and soft power for political transition or change instead of conflict. The third was a "bottom-up" instead of a "top-down" approach for social change. He implemented the last through scholarly works and educating people.

## 7. Conclusions

The Hudaybiyya peace treaty is one of the turning points in Islamic history. It is the most significant treaty of the prophetic government (Khorramshad and Mojtahedi 2020). The atmosphere of peace created by this pact allowed the rapid spread of Islam in the Arabian Peninsula. After analyzing its significance, this article focused on how ʿAbd Allah Ibn ʿUmar, a leading companion of the Prophet, deeply comprehended this peace treaty

for peaceful coexistence. Understanding the essence of this treaty, he made every effort to put it into practice during the first and second *fitan* periods in the history of Islam, but was unsuccessful. Instead of fighting or using force against injustice, he preferred diplomacy and soft power between rivals. In addition, he enlightened intellectuals and ordinary people through his scholarly works. Later, the principles of peace that he put forward had a great impact on Islamic scholars. History testifies that Ibn ʿUmar's approach to civil wars was relevant for a peaceful society and social cohesion. This article concludes by mentioning a coincidence that happened to Ibn ʿUmar—it may be an additional sign of the Ḥudaybiyya treaty's impact on his role. Ibn ʿUmar was the first to make allegiance in Ḥudaybiyya and the only one to take it twice. Coincidentally, he was the first Muslim, next to the Prophet, who, through his in-depth understanding of *sīrah*, discovered the practical implications of this pact in retaining peace. Similarly, *ḥajj* days came twice when wars were in progress, and on both occasions, Ibn ʿUmar mentioned the Ḥudaybiyya treaty.

**Author Contributions:** Mursal Farman extracted a subchapter of his PhD thesis *"Examining Ibn ʿUmar's Stance during Fitan Times and its Impact: Re-reading His Approaches to Peace and Conflict"* and wrote the original draft of the article. Salih Yucel added some contemporary Turkish and English sources and explained the importance of Hudaybiyya treaty. All authors have read and agreed to the published version of the manuscript.

**Funding:** This research received no external funding.

**Conflicts of Interest:** The authors declare no conflict of interest.

## Notes

[1] According to another tradition, Abū Sinān al-Asadī (d. 5/627) was the first one to take pledge at Ḥudaybiyya (Ibn Abī Shaybah 1989, no. 32508).

[2] To complete ʿ*umrah*, men must shave/trim the hair on their heads, while women should cut it short, to the length of their fingertips.

[3] The English translation of this *ḥadīth* was taken from the following website: Sahih Muslim 1301c—The Book of Pilgrimage—كتاب الحج—Sunnah.com—Sayings and Teachings of Prophet Muhammad صلى الله عليه و سلم

[4] Here, a point may be raised that there were many other companions who died after Ibn ʿUmar, e.g., Jābir b. ʿAbd Allah (d. 687), Anas b. Mālik (d. 709), Abū Ṭufayl ʿĀamir b. Wāthila (d. 721), or numerous other companions who were Ibn ʿUmar's peers, such as Ibn ʿAbbās (d. 687), etc. There is a very clear-cut, obvious answer to this objection—of the companions who passed away after Ibn ʿUmar, such as Jābir, Anas, and Abū Ṭufayl, none were Qurayshi émigré. They neither had any experience with the Prophet in Mecca nor were relatively influential. As far as Ibn ʿAbbās is concerned, he was born just three years before the Prophet's migration. Therefore, his migration and acceptance of Islam occurred a little earlier, before the conquest of Mecca. In addition, in the early days of the second wave of unrest, he had already lost his eyesight and hence could not pass a very active life in his later years. Moreover, he breathed his last during the second wave of unrest. Likewise, the companion ʿAmr b. Ḥurayth (d. 705) happened to be born during the days of the battle of Badr and was among the junior companions. Therefore, these companions were not focused on in the subject under discussion.

[5] Ibn ʿUmar participated in both Muʿāwiya's peace treaties: with Caliph ʿAlī in 657 and with Imām Ḥasan in 661. One of these was on the counsel of his sister, Ḥafṣa. Scholars differ on which one of these two occasions Ḥafṣa advised him to partake in.

[6] Such as Ibn al-Zubayr, ʿAbd Allāh b. Ḥanẓala (d. 683), Ibn Muṭīʿ (d. 629), etc.

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
