# Peer review of "Rereading the Hudaybiyya Treaty: With Special Reference to Ibn ʿUmar’s Role in Fitan†"

_religions, doi:10.3390/rel14050666_

Round 1
Reviewer 1 Report
Would substitute 'critical moment' or 'turning point' for 'brilliant' in the abstract.
The article is well researched and cogently argued. It sheds new light on the implication for peace-making of the Treaty of Hudaybiyya and 'Umar's subsequent role, and it certainly merits publishing. However, literature on the topic of peace and war and Islamic states often cites the Treaty's ten year validity as evidence that Muslims can only enter temporary truces, and that a permanent state of war exists between the Muslim and non-Muslim worlds. See https://www.google.com/books/edition/Proposed_Arms_Sales_to_Jordan/zJ2E3yVhgvQC?hl=en&gbpv=1&dq=can+muslims+states+only+make+peace+treaties+for+then+years&pg=PA68&printsec=frontcover.
In my opinion, a response to this argument needs to be added to the paper.
Author Response
We looked at the source provided by the reviewer, but it is about arms sales to Jordan and records the discussion of members of the Senate about the representative in the US Congress on this issue.
However, we added “The period of time of peace treaty should be considered in its context. It does not mean Muslims can only make peace treaties for ten years” (p 4). We thank the reviewer for this point.
Reviewer 2 Report
It's a bit short and doesn't quite engage with the literature in the field.
Author Response
The length of the article is 8485 words. Yes, the topic can be a PhD topic. But we think over 8000 words will be enough for such an article. However, we added about 450 words to the article.
Reviewer 3 Report
I have added some comments here and there. I strongly suggest to insert translations in all the cases that arab terms that are not of common knoledge are mentioned.
An excursus about fitan and its historical is necessary
Author Response
It is a relevant point, and we thank the reviewer for this. The translation of the Arabic words is done when they are first time mentioned in the article. That is the methodology of academic writing.
Reviewer 4 Report
While there is some merit to research, it needs significant work before it can be published. The most specific issues are below, but without these being addressed it is difficult to review the article more thoroughly as there is not enough reference information to adequately judge the merits of the article. There also seems to be an Arabic language article on the same topic. I do not have access to this article but the reference for it is Housayni Khalid and Dafroor Raabih, published in Majallah al-Ihya' (v 20 no 3, Sept 22, 2020; pg 481-500)
-Section 3 The Ḥudaybiyya Treaty and Ibn ʿUmar’s Role is too speculative for an academic article. This section should be completely rewritten to only include factually verifiable information.
-Line 188-9 "Ibn ʿUmar came to the Prophet and became the first person to pledge allegiance." This information is inaccurate.
Line 292 How can there be thousands of casualties when only at most a few hundred people took part in the wars? I could find no reference for this claim.
-Specific Hadith numbers are missing and these should all be included. This can be done either in AJ Winsick style or hadith number. But it is a must to have the specific hadith referenced.
-Beyond missing hadith specific hadith numbers, there are too many other instances of missing information in the references, e.g., there are no page numbers for any references. Additional incomplete information in some notes includes e.g. in no 41 where it says accessed on, but there is no date.
-Some of the sources used are not good sources to use for a journal article. These include The Encyclopedia Britannica and Fredrick Denny's Oxford Bibliography article. These should be changed to a proper source.
-Armstrong 2006 is listed several times in the text of the article, but this is not found in the References section
Author Response
Our response point by point is in bold.
While there is some merit to research, it needs significant work before it can be published. The most specific issues are below, but without these being addressed it is difficult to review the article more thoroughly as there is not enough reference information to adequately judge the merits of the article. There also seems to be an Arabic-language article on the same topic.
It is a relevant point. We discussed it a bit further and added 7 English, 3 Turkish and one Arabic contemporary sources on Hudaybiyya.
I do not have access to this article but the reference for it is Housayni Khalid and Dafroor Raabih, published in Majallah al-Ihya' (v 20 no 3, Sept 22, 2020; pg 481-500)
We have gone through the article مواقف عبد الله بن عمر من الفتن السياسية في عصره (Ibn ʿUmar’s perspective of political strife of his time). The article is good but depicts a very simple picture of Ibn ʿUmar. Our research takes a completely different approach. The authors mention that Ibn ʿUmar learnt from the Prophet not to shed blood of the Muslims. We agree with this point and added to the article (see p 11). However we add more elements that played important role in the formation of Ibn ʿUmar’s personality and led him to assume this direction during fitan period.
-Section 3 The Ḥudaybiyya Treaty and Ibn ʿUmar’s Role is too speculative for an academic article. This section should be completely rewritten to only include factually verifiable information.
The word “too speculative” is not justice especially in a situation where cause and effect both exist but need some speculation to get connected. We have referred to the original sources while discussing Ibn ʿUmar’s personality development, distinct obedience of the prophet, etc. learning during Hudaybiyya, Ibn ʿUmar’s relationship with the late converts (like Khalid b. al-Walid) after Hudaybiyya, his balanced opinion about them after a dispute over their leadership between his father ‘Umar and Abu Bakr and then his balanced perspective of the leadership of late converts during fitan wars especially when he refers to Hudaybiyya. In the presence of all these facts, a researcher needs to be a little speculative.
-Line 188-9 "Ibn ʿUmar came to the Prophet and became the first person to pledge allegiance." This information is inaccurate.
This is not our argument but still it is based on historical sources. Our reference went missing in the previous version after the article was pasted into the template of the journal. We have now added the missing source (al-Ma'arif by Ibn Qutayba al-Dinawari) See p 5. Ibn Qutayba clearly mentions it in his book (p. 162):
وكان أوّل من بايع عبد الله بن عمر، وكانت البيعة بسبب عثمان بن عفان،- رضى الله عنه- وذلك أنه بعثه إلى مكة ليخبر قريشا أنه لم يأت لحرب، فاحتبسته. «قريش» عندها، وبلغ رسول الله- صلّى الله عليه وسلم- أنه قد قتل. فدعا الناس إلى البيعة على مناجزة القوم، ثم بلغه أن الّذي ذكر في أمر «عثمان» باطل.
There is another opinion too that Abū Sinān al-Asadī (d. 5/627) was the first one to take pledge at Ḥudaybiya (Ibn Abī Shayba 1989, no. 32508). We have added it in the article. See (p. 5)
Line 292 How can there be thousands of casualties when only at most a few hundred people took part in the wars? I could find no reference for this claim.
The reviewer has misunderstood our statement. There were no doubt thousands of casualties in fitan wars, but the participants were mostly non-ṣaḥāba. Ṣaḥāba’s participation was very limited as has been mentioned in the article. I will only give one example of the battle of the Camel by quoting a paragraph from al-Tabari (V, 539): He mentions ten thousand casualties only in this battle.
كَانَ قتلى الجمل حول الجمل عشرة آلاف، نصفهم من أَصْحَاب علي، ونصفهم من أَصْحَاب عَائِشَة، من الأزد ألفان، ومن سائر اليمن خمسمائة، ومن مضر الفان، وخمسمائة من قيس، وخمسمائة من تميم، والف من بنى ضبة، وخمسمائة من بكر بن وائل وقيل: قتل من أهل الْبَصْرَة فِي المعركة الأولى خمسة آلاف، وقتل من أهل الْبَصْرَة فِي المعركة الثانية خمسة آلاف، فذلك عشرة آلاف قتيل من أهل الْبَصْرَة، ومن أهل الْكُوفَة خمسة آلاف.
-Specific Hadith numbers are missing and these should all be included. This can be done either in AJ Winsick style or hadith number. But it is a must to have the specific hadith referenced.
-Beyond missing hadith specific hadith numbers, there are too many other instances of missing information in the references, e.g., there are no page numbers for any references. Additional incomplete information in some notes includes e.g. in no 41 where it says accessed on, but there is no date.
We had referenced our sources according to APA style. Page numbers are not given in the referencing section. See https://libguides.library.usyd.edu.au/citation/apa7. However, the required missing details have been included in the current version of the article.
-Some of the sources used are not good sources to use for a journal article. These include The Encyclopedia Britannica and Fredrick Denny's Oxford Bibliography article. These should be changed to a proper source.
We disagree with the reviewer. Many other academic sources used Encyclopedia Britannica and Fredrick Denny's Oxford Bibliography article.
-Armstrong 2006 is listed several times in the text of the article, but this is not found in the References section
This is done. Thanks for that.
Reviewer 5 Report
The article is interesting. It has an important subject. The Hudaybiyya treaty, a turning point in Islamic history. However, the article follows what I could explain as a strictly pro-Islamic perspective. I do respect the author to entertain such a perspective. However, as it is, the article sounds highly subjective. In consequence, it appears more like a didactical text. Secondly, the article, which should be appreciated for incorporating classical resources, is however weak in terms of referencing contemporary studies in this regard. The author(s) should have use relevant studies (both on Hudaybiyya or early Islamic period) at least to contextualise its subject within a consistent scholarly framework.
Author Response
Our response point by point is in bold.
The article is interesting. It has an important subject. The Hudaybiyya treaty, a turning point in Islamic history. However, the article follows what I could explain as a strictly pro-Islamic perspective. I do respect the author to entertain such a perspective. However, as it is, the article sounds highly subjective. In consequence, it appears more like a didactical text.
Our explanation is based on Arabic historical sources. Regrettably, there are no other sources or non-Muslim historical sources. If we go against historical Arabic sources, the field experts will harshly criticise and question academic reliability. However, we provided various opinions.
Secondly, the article, which should be appreciated for incorporating classical resources, is however weak in terms of referencing contemporary studies in this regard. The author(s) should have use relevant studies (both on Hudaybiyya or the early Islamic period) at least to contextualise its subject within a consistent scholarly framework.
We thank to the reviewer. This is a relevant point. We added eleven contemporary sources about the Hudaybiyya.
Round 2
Reviewer 4 Report
Overall the additions make the article ready for publication. Before final publication, however, the authors should better integrate the new material into the text and have more analysis of the new material.
Author Response
Thanks for the comments. They really helped us in improving the article.
Reviewer 5 Report
I am now satisfied with the revision. I have no longer any reservation.
Author Response
Thank you for the constructive comments